

# Recognition of apnea and hypopnea by non-contact optical fiber mattress and its application in the diagnosis of obstructive sleep apnea hypopnea syndrome: a retrospective study

Ling Zhou[1],*, Xiaoyan Zhu[1],*, Lu Liu[1], Lingling Wang[1],
Pengdou Zheng[1], Fengqin Zhang[1], Zhenyu Mao[1], Guoliang Huang[2],
Songlin Cheng[2], Huiguo Liu[1] and Wei Liu[3]

[1] Department of Respiratory and Critical Care Medicine, Tongji Hospital, Tongji Medical College, Huazhong University of Science and Technology, Wuhan, Hubei, China
[2] Center for Intelligent Optoelectronics, Wuhan University of Technology, Wuhan, Hubei, China
[3] Department of Geriatrics, Tongji Hospital, Tongji Medical College, Huazhong University of Science and Technology, Wuhan, Hubei, China
* These authors contributed equally to this work.

Corresponding author
Wei Liu, 404793938@tjh.tjmu.edu.cn

## ABSTRACT

**Objectives:** This study sought to evaluate the diagnostic value of a non-contact optical fiber mattress for apnea and hypopnea and compare it with traditional polysomnography (PSG) in adult obstructive sleep apnea hypopnea syndrome (OSAHS).

**Methods:** To determine the value of a non-contact optical fiber mattress for apnea and hypopnea, six healthy people and six OSAHS patients were selected from Tongji Hospital to design a program to identify apnea or hypopnea. A total of 108 patients who received polysomnography for drowsiness, snoring or other suspected OSAHS symptoms. All 108 patients were monitored with both the non-contact optical fiber mattress and PSG were collected.

**Results:** Six healthy controls and six patients with OSAHS were included. The mean apnea of the six healthy controls was 1.22 times/h, and the mean hypopnea of the six healthy controls was 2 times/h. Of the six patients with OSAHS, the mean apnea was 12.63 times/h, and the mean hypopnea was 19.25 times/h. The non-contact optical fiber mattress results showed that the mean apnea of the control group was 3.17 times/h and the mean hypopnea of the control group was 3.83 times/h, while the mean apnea of the OSAHS group was 11.95 times/h and the mean hypopnea of the OSAHS group was 17.77 times/h. The apnea index of the non-contact optical fiber mattress was positively correlated with the apnea index of the PSG ($P < 0.05$, r = 0.835), and the hypopnea index of the non-contact optical fiber mattress was also positively correlated with the hypopnea index of the PSG ($P < 0.05$, r = 0.959). The non-contact optical fiber mattress had high accuracy (area under curve, AUC = 0.889), specificity (83.4%) and sensitivity (83.3%) for the diagnosis of apnea. The non-contact fiber-optic mattress also had high accuracy (AUC = 0.944), specificity (83.4%) and sensitivity (100%) for the diagnosis of hypopnea. Among the 108 patients enrolled, there was no significant difference between the non-contact

optical fiber mattress and the polysomnography monitor in total recording time, apnea hypopnea index (AHI), average heart rate, tachycardia index, bradycardia index, longest time of apnea, average time of apnea, longest time of hypopnea, average time of hypopnea, percentage of total apnea time in total sleep time and percentage of total hypopnea time in total sleep time. The AHI value of the non-contact optical fiber mattress was positively correlated with the AHI value of the PSG ($P < 0.05$, r = 0.713). The specificity and sensitivity of the non-contact optical fiber mattress AHI in the diagnosis of OSAHS were 95% and 93%, with a high OSAHS diagnostic accuracy (AUC = 0.984).

**Conclusion:** The efficacy of the non-contact optical fiber mattress for OSAHS monitoring was not significantly different than PSG monitoring. The specificity of the non-contact optical mattress for diagnosing OSAHS was 95% and its sensitivity was 93%, with a high OSAHS diagnostic accuracy.

## INTRODUCTION

Obstructive sleep apnea hypopnea syndrome (OSAHS) is characterized by apnea or hypopnea during sleep (*Gottlieb, 2022*). Apnea and hypopnea result in intermittent hypoxia and sleep fragmentation. Untreated OSAHS is associated with significant decreases in quality of life and cognitive performance, increased risk of accidents due to daytime sleepiness, as well as a range of other complications (*Tete et al., 2023*). OSAHS is an independent risk factor for hypertension, diabetes and coronary heart disease (*Zeng et al., 2022*, *Wilson, Veatch & Johnson, 2022*, *Baker-Smith et al., 2021*), increasing risk of overall mortality by 26.2%. OSAHS and its cardiovascular or metabolic sequelae may be caused by intermittent hypoxemia-induced oxidative stress and inflammation, as well as increased sympathetic activation (*Wang et al., 2022*). Nocturnal polysomnography (PSG) monitoring is the current gold standard for diagnosing OSAHS (*Hu et al., 2022*; *Karabul et al., 2022*). However, PSG monitoring has many shortcomings: it is expensive, laborious and requires a full night of sleep monitoring in the hospital. The presence of numerous wearable sensors is also highly inconvenient and can reduce sleep quality. Most OSAHS patients are not diagnosed in a timely manner due to the expensive and time-consuming monitoring needed. Therefore, the development of a simple, comfortable, and convenient diagnostic screening device for OSAHS would likely improve OSAHS diagnostic rates (*Younes et al., 2022*).

The non-contact optical fiber mattress (non-contact vital signs monitor AVS-1, Wuhan Kairuipu Medical Technology Co., LTD, Wuhan, China) is a new, non-wearable device that is placed under any mattress and can sense respiratory motion pressure, torso status, heart rate, and snoring using optical fiber supported by a deep learning algorithm. The mattress records a series of indicators reflecting sleep status, such as the apnea

hypopnea index (AHI), that can be obtained on a smartphone application. The greatest advantage of this device is that patients suspected of OSAHS can comfortably complete diagnostic testing in their own bed at home without being constrained by any sensors. The device also has the benefits of non-inductive monitoring, intelligent data and the capability for long-term monitoring. This study evaluated the diagnostic performance of the non-contact optical fiber mattress for OSAHS and compared patients with suspected OSAHS with PSG monitoring along with non-contact optical fiber mattress monitoring to assess the accuracy and effectiveness of the non-contact optical fiber mattress.

## MATERIALS AND METHODS

### Study subjects

To explore the efficacy of the non-contact optical fiber mattress in determining apnea and hypopnea, this study first recruited six healthy controls and six patients with a clear diagnosis of OSAHS by PSG monitoring from November 2021 to December 2021 at Tongji Hospital, Tongji Medical College, Huazhong University of Science and Technology (the Ethics Committee of Tongji Hospital, Tongji Medical College, Huazhong University of Science and Technology granted ethical approval to carry out the study within its facilities, IRB approval number: TJ-IRB20221307). Written informed consent was collected from all study participants. All 12 patients were monitored during one night of sleep using both a non-contact optical fiber mattress (non-contact vital signs monitor AVS-1, Wuhan Kairuipu Medical Technology Co., LTD; Fig. 1) and a PSG polysomnography monitor (SOMNO screen plus PSG+). The respiratory motion waveform signal data from the non-contact optical fiber mattress were compared with the simultaneously-recorded PSG signal data. The respiratory waveform data were also compared between the healthy controls and the OSAHS patients to identify apnea or hypopnea respiratory waveform characteristics, which were then combined with graphic recognition technology and big data processing to design an intelligent program to identify apnea or hypopnea.

To evaluate the diagnostic efficacy of the non-contact optical fiber mattress in adults with OSAHS, 108 patients who received sleep monitoring for drowsiness, snoring, or other suspected OSA symptoms at Tongji Hospital, Tongji Medical College, Huazhong University of Science and Technology from January 2022 to September 2022 were recruited for this study. All participants underwent sleep monitoring using both a non-contact optical fiber mattress and PSG polysomnography. This study was approved by the Ethics Committee of Tongji Hospital, Tongji Medical College, Huazhong University of Science and Technology. Written informed consent was also collected from all study participants.

Inclusion criteria: patients with a complete medical history, demographic data, relevant examinations and sleep monitoring data.

Exclusion criteria: patients with psychiatric disorders, pregnant women, and patients with episodic sleep disorders, sleep-related motor diseases and other sleep-related diseases. All data were recorded after a standardized medical history and physical examination by a specialist physician.

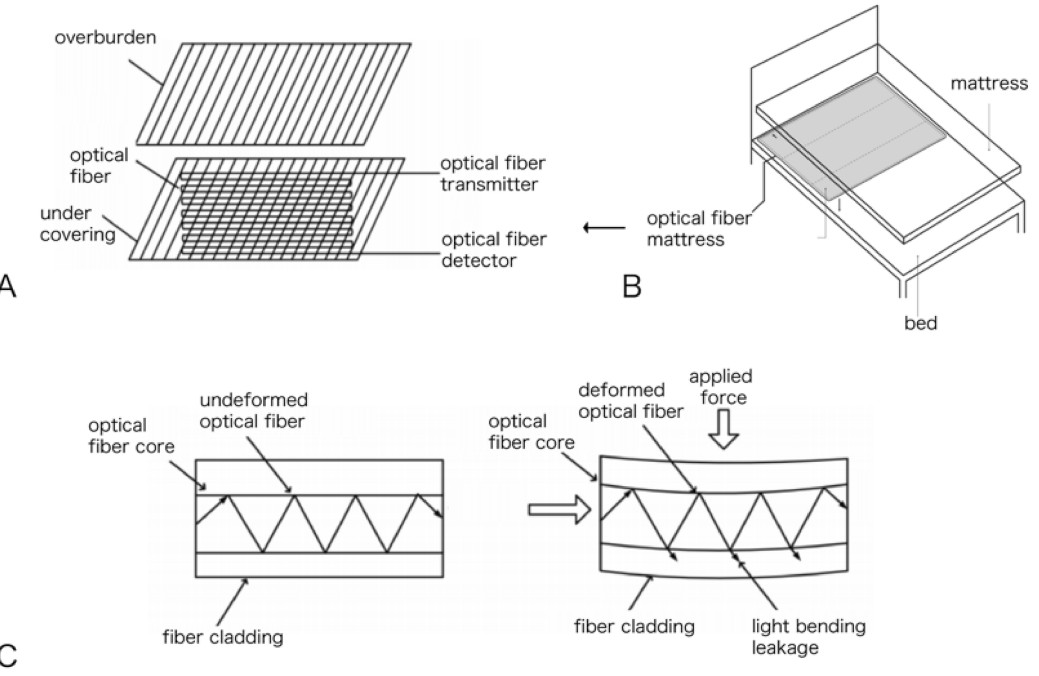

**Figure 1 Non-contact optical fiber mattress AVS-1: (A) optical fiber mattress structure diagram; (B) optical fiber mattress placement diagram; (C) transmission loss diagram of microcurved fiber.**

### General data collection

General patient data, including gender, age, body mass index (BMI), neck circumference, waist circumference, and diagnostic information on hypertension, diabetes mellitus, coronary artery disease and gastroesophageal reflux, were collected.

### Sleep monitoring related test indicators

Total recording time: Total PSG recording time was the total time between when the patient put on the PSG monitoring device and when the patient removed the PSG device; the total non-contact optical fiber mattress recording time was the total time between when the patient lay on the mattress and prepared to sleep and when the patient got up and left the mattress.

Total sleep time: For both the PSG and non-contact optical fiber mattress, total sleep time was based on dynamic measurements of heart rate, respiration and movement. When the patient's heart rate and respiration were stable and body movement decreases lasted for several min, the patient was classified as sleeping. The sleeping period ended when the patient's body movement increased, their heart rate and respiration fluctuated violently or they got up and left the mattress. Several periods of sleep could occur during the whole night of sleep monitoring; all periods of sleep were added together and recorded as the total sleep time.

AHI: AHI was calculated by dividing the total number of apnea and hypopnea events by total recording time in hours. Apnea and hypopnea were both defined using respiratory movement measurements with the non-contact optical fiber mattress: a respiratory wave

amplitude 20% of the normal respiratory wave lasting more than 10 s was considered an apnea, and a respiratory wave amplitude 40% of the normal respiratory wave lasting more than 10 s was considered a hypopnea.

Average heart rate: The average heart rate of the total recording time.

Tachycardia index: The number of times that the average heart rate exceeded 100 beats/min per h during the total recording time (not less than 7 h).

Bradycardia index: The number of times that the average heart rate was lower than 60 beats/min per h during the total recording time (not less than 7 h).

Average respiratory rate: Average respiratory rate of the total recording time.

Body movement index: Average number of times patient turned over or got up per hour during the total recording time.

Maximum time of apnea: The longest apnea event during the total recording time.

Mean time of apnea: Total apnea time (calculated by adding together the time of each apnea event) divided by the number of apnea events.

Maximum time of hypopnea: The longest hypopnea event during the total recording time.

Mean time of hypopnea: Total hypopnea time (calculated by adding together the time of each hypopnea event) divided by the number of hypopnea events.

Percentage of total apnea time in total sleep time: The total apnea time divided by the total sleep time.

Percentage of total hypopnea time in total sleep time: The total hypopnea time divided by the total sleep time.

## Statistical analysis

Prism 8.0 was used for statistical analysis: $n$ (%) was used for qualitative data, mean ± standard deviation was used for quantitative data of normal distribution and median of interquartile interval was used for quantitative data of non-normal distribution. Pearson analysis was used for the correlation analysis, the correlation coefficient was expressed as r and ROC curve was used for evaluating diagnostic performance. $P < 0.05$ was considered statistically significant.

## RESULTS

### Non-contact optical fiber mattress waveform diagram of different breathing movements and the corresponding SPO$_2$ by PSG

The non-contact optical fiber mattress senses different respiratory motion pressure conditions using fiber optics. The respiratory motion waveform signals of the non-contact optical fiber mattress were recorded, collected and analyzed. The normal respiratory motion waveform is shown as a regular and uniform respiratory motion diagram in Fig. 2A. There was no obvious fluctuation in SPO$_2$ of normal respiratory movement under PSG monitoring during the corresponding time period (Fig. 2B). Obstructive sleep apnea manifested as a spindle-shaped breathing movement diagram in which the breathing movement first changed from large to small until apnea and then changed from small to large (Fig. 2C). SPO$_2$ reduction was observed in obstructive sleep apnea under PSG

monitoring during the corresponding time period (Fig. 2D). Central sleep apnea manifested as a sudden cessation of respiratory movement and then a semi-spindle breathing movement chart that changed from small to large (Fig. 2E). $SPO_2$ reduction was also observed in central sleep apnea under PSG monitoring during the corresponding time period (Fig. 2F). Hypopnea manifested as a marked reduction in respiratory movement, though respiratory movement did not completely stop (Fig. 2G). $SPO_2$ of hypopnea decreased in stages under PSG monitoring during the corresponding period (Fig. 2H).

## Non-contact optical fiber mattress and PSG for detecting AHI in patients

Six healthy controls with a mean age of 43.67 years and six patients with OSAHS and a mean age of 44.33 years were included in the analysis. After PSG monitoring throughout the night, the mean apnea of the six healthy controls was 1.22 times/h, and the mean hypopnea of the six healthy controls was 2 times/h. Of the six patients with OSAHS, the mean apnea was 12.63 times/h, and the mean hypopnea was 19.25 times/h. The respiratory motion waveform signal data collected by the non-contact optical fiber mattress were analyzed and compared with the synchronous PSG signal data to determine the respiratory motion signal characteristics of apnea and hypopnea and to design an intelligent program to judge the criteria through relevant algorithms. The non-contact optical fiber mattress results showed that the mean apnea of the control group was 3.17 times/h and the mean hypopnea of the control group was 3.83 times/h, while the mean apnea of the OSAHS group was 11.95 times/h and the mean hypopnea of the OSAHS group was 17.77 times/h (Table 1).

## Effectiveness of the non-contact optical fiber mattress in determining apnea and hypopnea

To investigate the correlation between the non-contact optical fiber mattress and PSG monitoring results, the AHI values of the non-contact optical fiber mattress were correlated with the apnea index and hypopnea index recorded by the PSG. The apnea index of the non-contact optical fiber mattress was positively correlated with the apnea index of the PSG (Fig. 3A), with a correlation coefficient r = 0.835 ($P < 0.05$). The hypopnea index of the non-contact optical fiber mattress was positively correlated with the hypopnea index of the PSG (Fig. 3B), with a correlation coefficient r = 0.959 ($P < 0.05$). ROC curves were then plotted using the apnea index and hypopnea index of the non-contact optical fiber mattress. The results suggest that the non-contact optical fiber mattress is valuable for determining both apnea and hypopnea ($P < 0.05$). The non-contact optical fiber mattress was highly accurate in diagnosing apnea, with an area under the curve (AUC) of 0.889, a specificity of 83.4% and a sensitivity of 83.3% when taking the maximum of the Youden index (Fig. 3C). The non-contact optical fiber mattress also had high accuracy in diagnosing hypopnea, with an AUC of 0.944, a specificity of 83.4% and a sensitivity of 100% when taking the maximum of the Youden index (Fig. 3D).

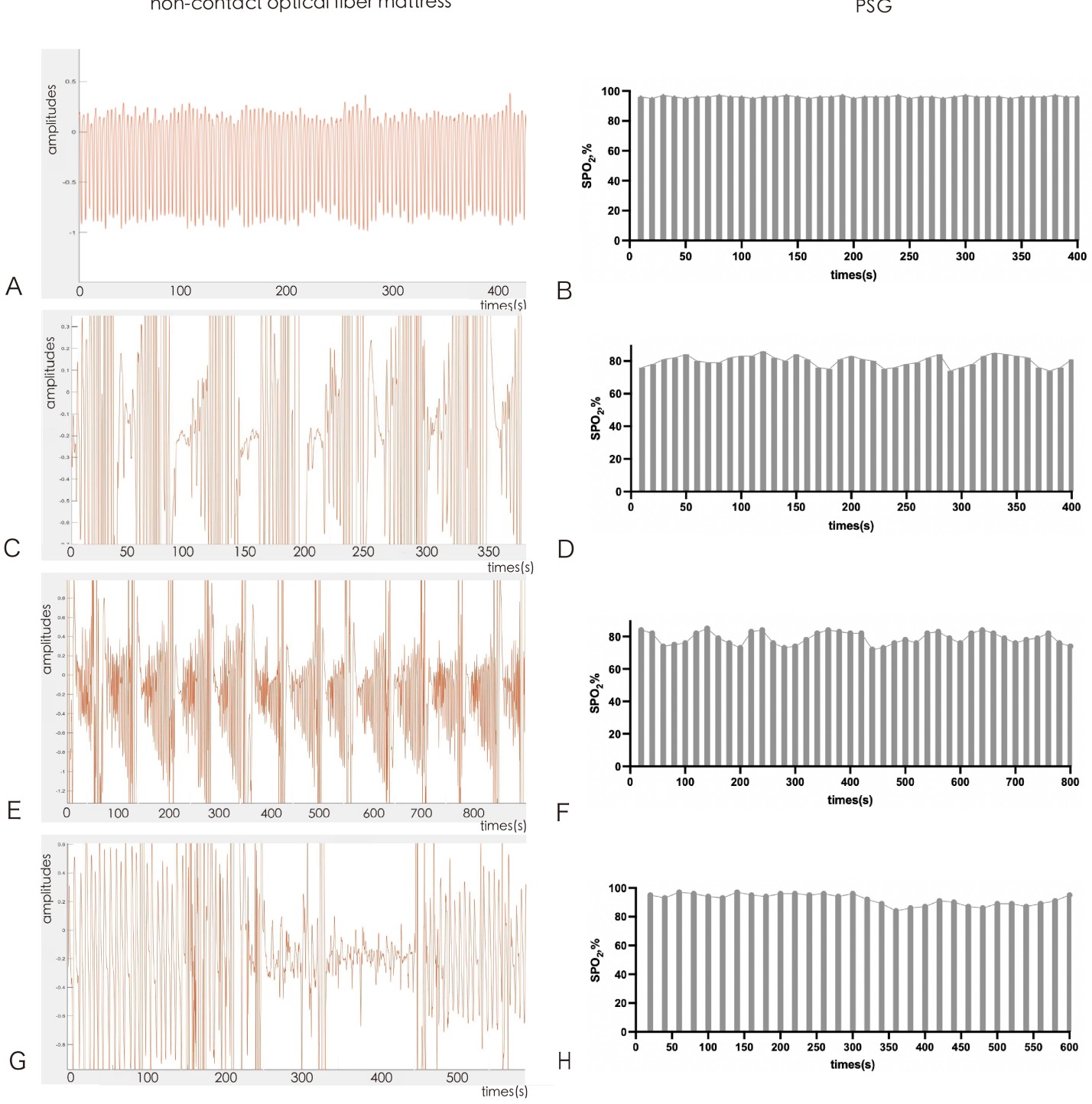

**Figure 2 Non-contact optical fiber mattress waveform diagram: (A and B) normal respiratory motion waveform; (C and D) obstructive sleep apnea waveform; (E and F) central sleep apnea waveform; (G and H) hypopnea waveform.**

**Table 1  Non-contact optical fiber mattress and PSG for detecting AHI ($n = 6$).**

|  | PSG | | | ANVENUS SLEEP MONITOR | | |
|---|---|---|---|---|---|---|
| Subjects | A | H | A+H | A | H | A+H |
| Control |  |  |  |  |  |  |
| 1 | 0.4 | 1.4 | 1.8 | 0.6 | 3.4 | 4 |
| 2 | 1.4 | 3.5 | 4.9 | 1.5 | 7.6 | 9.1 |
| 3 | 0.6 | 10 | 10.6 | 2.4 | 3.2 | 5.6 |
| 4 | 0.8 | 1.8 | 2.6 | 3.3 | 2.4 | 5.7 |
| 5 | 2.5 | 1.9 | 4.4 | 1.4 | 2.8 | 4.2 |
| 6 | 1.6 | 0.4 | 2 | 2.8 | 3.6 | 6.4 |
| OSAHS |  |  |  |  |  |  |
| 1 | 8.5 | 3.2 | 11.7 | 12.4 | 5.4 | 17.8 |
| 2 | 12.3 | 17.4 | 29.7 | 11.4 | 12.4 | 23.8 |
| 3 | 9.5 | 9.1 | 18.6 | 8.5 | 6.7 | 15.2 |
| 4 | 8.5 | 6.8 | 15.3 | 21.5 | 24.5 | 46 |
| 5 | 21.4 | 17.5 | 38.9 | 43.2 | 37.5 | 80.7 |
| 6 | 15.6 | 17.7 | 33.3 | 18.5 | 20.1 | 38.6 |

Note:
A, Apnea index per h; H, hypopnea index per h; OSAHS, obstructive sleep apnea hypopnea syndrome.

## Analysis of general patient information

Among the 108 patients who underwent sleep monitoring in this study, 84 patients (77.8%) were male and 24 patients (22.2%) were female. The average age of these participants was 47.75 years, the average BMI was 26.47 kg/m$^2$, the average neck circumference was 37.37 cm, the average waist circumference was 84.40 cm, the average history of snoring was 19.55 years, 34 patients had hypertension (31.5%), 19 had diabetes (17.6%), 16 had coronary heart disease (14.8%) and nine had gastroesophageal reflux (8.3%; Table 2).

## Non-contact optical fiber mattress and PSG sleep monitoring index comparison analysis

After verifying the high accuracy of the non-contact optical fiber mattress for both apnea and hypopnea, this study further evaluated the diagnostic accuracy of the non-contact optical fiber mattress for OSAHS in the 108 patients that were monitored with both the non-contact optical fiber mattress and the polysomnography monitor. The indexes of the two monitoring systems were compared and the comparison showed that there was no difference in total recording time and apnea index between the two groups. However, the hypopnea index of the non-contact optical fiber mattress group was lower than that of the PSG group, and the difference was statistically significant ($P < 0.05$). AHI was calculated by adding the apnea index and the hypopnea index and the results showed that in the range of AHI < 5, there was a statistically significant difference ($P < 0.05$) between the AHI values of the non-contact optical fiber mattress group and the PSG group; however, for the diagnosis of mild obstructive sleep apnea ($5 \leq$ AHI < 15), moderate obstructive sleep apnea ($15 \leq$

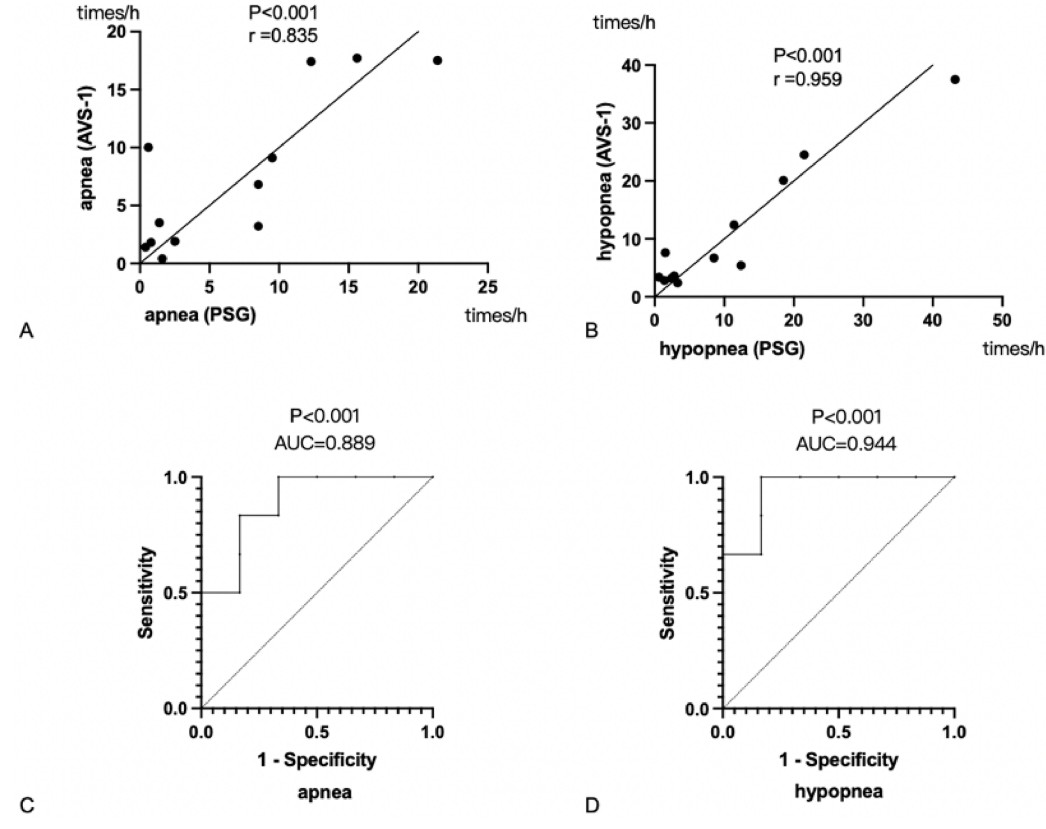

**Figure 3 Correlation and ROC analysis.** (A) Correlation analysis of apnea index between non-contact optical fiber mattress AVS-1 and PSG. (B) Correlation analysis of low ventilation index between non--contact optical fiber mattress and PSG. (C) The value of the non-contact optical fiber mattress for determining apnea. (D) The value of the non-contact optical fiber mattress for determining hypopnea. PSG: Polysomnography. The correlation coefficient is expressed as r: the larger the r value, the greater the correlation. AUC, the area under the curve; an AUC value of 0.5–0.7 indicates a low accuracy, an AUC value of 0.7–0.9 indicates a medium accuracy and an AUC value greater than 0.9 indicates a high accuracy.

AHI < 30) and severe obstructive sleep apnea (AHI ≥ 30), there was no significant difference in AHI values between the two groups. The non-contact optical fiber mattress group had a significantly higher somatic motion index than the PSG group, and the difference was statistically significant ($P < 0.05$). In contrast, there were no significant differences between the two groups in mean heart rate, tachycardia index, bradycardia index, maximum time of apnea, mean time of apnea, maximum time of hypopnea, mean time of hypopnea, total time of apnea as a percentage of total sleep time and total time of hypopnea as percentage of total sleep time (Table 3).

## Correlation analysis between non-contact optical fiber mattress and PSG

In order to explore the correlation between non-contact optical fiber mattress and PSG monitoring, the AHI value of the non-contact optical fiber mattress and the corresponding AHI value of the PSG group were analyzed. The results showed that the AHI value of the

**Table 2** Analysis of general information (*n* = 108).

| Items | *n* | % |
|---|---|---|
| Age, years | 47.75 ± 11.76 | – |
| BMI, kg/m² | 26.47 ± 3.91 | – |
| Neck circumference, cm | 37.37 ± 2.55 | – |
| Waist circumference, cm | 84.40 ± 3.25 | – |
| History of snoring, years | 19.55 ± 14.16 | – |
| Gender | | |
| Male | 84 | 77.8 |
| Female | 24 | 22.2 |
| Hypertension | 34 | 31.5 |
| Diabetes | 19 | 17.6 |
| Coronary heart disease | 16 | 14.8 |
| Gastroesophageal reflux | 9 | 8.3 |

**Note:**
Qualitative data are expressed using *n* (%), and quantitative data with normal distribution are expressed as mean ± standard deviation. BMI, body mass index.

non-contact optical fiber mattress was positively correlated with the AHI value of the PSG, with a correlation coefficient r = 0.713 ($P < 0.05$; Fig. 4).

## Sensitivity and specificity of the non-contact optical fiber mattress for diagnosing OSAHS

PSG is the current gold standard for diagnosing OSAHS, with 100% sensitivity and specificity for diagnosis. However, the non-contact optical fiber mattress is more convenient and comfortable and less disruptive of sleep. In order to evaluate the efficacy of the non-contact optical fiber mattress for diagnosing OSAHS, the AHI of PSG was used as the basis for confirming OSAHS diagnosis and an ROC curve was drawn using the AHI index of the non-contact optical fiber mattress. The results suggest that the non-contact optical fiber mattress has diagnostic value for OSAHS ($P < 0.05$). The AHI of the non-contact optical fiber mattress had a high accuracy in diagnosing OSAHS, with an AUC of 0.984, a specificity of 95% and a sensitivity of 93% when the maximum value of the Youden index was taken (Fig. 5).

## DISCUSSION

After verifying the high accuracy of the non-contact fiber-optic mattress for both apnea and hypopnea in six healthy controls and six OSAHS patients, this study further compared the differences between the non-contact optical fiber mattress and PSG in assessing AHI and heart rate metrics. The evaluation results found that the hypopnea index was lower in the non-contact optical fiber mattress group than in the PSG group. Also, the AHI values monitored by the non-contact optical fiber mattress group differed to some extent from the AHI values in the PSG group, with statistically significant differences when AHI < 5. However, when used to diagnose mild obstructive sleep apnea, moderate obstructive sleep apnea and severe obstructive sleep apnea, respectively, there was no significant difference

**Table 3 Analysis sleep monitoring index of ANVENUS SLEEP MONITOR and PSG (*n* = 108).**

| Items | PSG | ANVENUS SLEEP MONITOR | Statistical values | *P* |
|---|---|---|---|---|
| Total recording time | 7.69 ± 0.69 | 7.84 ± 0.68 | 1.876 | 0.063 |
| Apnea, times/h | 13.46 ± 10.42 | 11.97 ± 9.26 | 0.714 | 0.477 |
| Hypopnea, times/h | 21.44 ± 17.36 | 17.95 ± 13.89 | 2.637 | 0.010* |
| AHI < 5 (*n* = 19) | 1.19 ± 1.09 | 2.45 ± 1.63 | 2.895 | 0.009* |
| 5 ≤ AHI < 15 (*n* = 14) | 8.26 ± 2.56 | 7.73 ± 1.26 | 0.411 | 0.687 |
| 15 ≤ AHI < 30 (*n* = 23) | 22.72 ± 4.02 | 24.18 ± 12.29 | 0.547 | 0.589 |
| AHI ≥ 30 (*n* = 52) | 54.79 ± 17.35 | 48.47 ± 16.45 | 1.824 | 0.074 |
| AHI | 32.50 ± 25.70 | 29.92 ± 23.15 | 1.435 | 0.154 |
| Average heart rate, times/min | 69.27 ± 9.75 | 67.47 ± 5.92 | 1.593 | 0.114 |
| Tachycardia index | 12.90 (4.13, 26) | 12 (5, 32.75) | 0.944 | 0.347 |
| Bradycardia index | 20.65 ± 18.06 | 23.66 ± 18.17 | 1.232 | 0.221 |
| Body movement index | 1.6 (1.3, 2.1) | 13 (4, 28) | 6.938 | 0.001* |
| Maximum time of apnea, s | 44.03 ± 30.31 | 37.61 ± 29.61 | 1.558 | 0.122 |
| Mean time of apnea, s | 21.39 ± 10.64 | 19.26 ± 10.41 | 1.490 | 1.139 |
| Maximum time of hypopnea,s | 79.94 ± 33.73 | 73.18 ± 43.12 | 1.914 | 0.583 |
| Mean time of hypopnea, s | 26.70 ± 9.69 | 30.55 ± 23.20 | 1.718 | 0.089 |
| Percentage of total apnea time in total sleep time, % | 0.66 (0.01, 2.21) | 1 (0.41, 5.21) | 1.765 | 0.080 |
| Percentage of total hypopnea time in total sleep time, % | 3.29 (1.34, 5.36) | 2.59 (0.45, 0.04) | 0.419 | 0.676 |

Notes:
Quantitative data with normal distribution were expressed as mean ± standard deviation. Quantitative data with non-normal distribution are expressed as median of interquartile spacing. AHI, apnea-hypopnea index.
* *P* < 0.05.

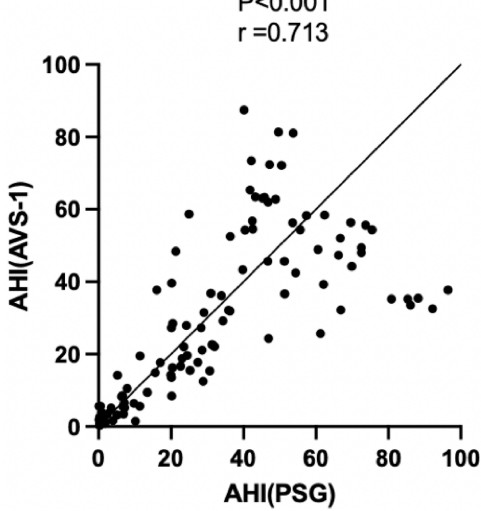

**Figure 4 Correlation analysis of AHI values between the non-contact optical fiber mattress AVS-1 and polysomnography (PSG).** The correlation coefficient is expressed as r, a larger r value indicates a greater correlation.

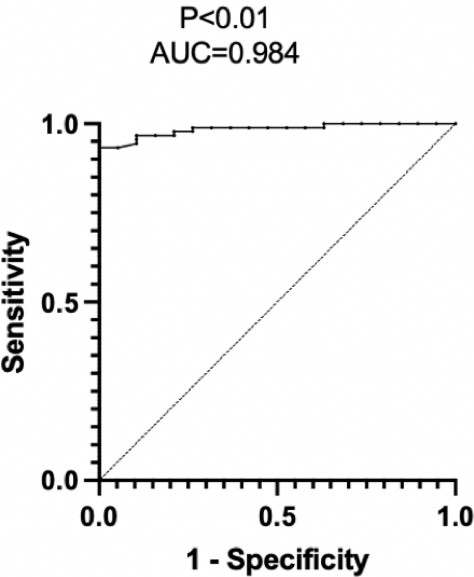

P<0.01
AUC=0.984

**Figure 5 Efficacy assessment of the non-contact optical fiber mattress AVS-1 for diagnosing OSAHS.** AUC, the area under the curve; an AUC value of 0.5–0.7 indicates a low accuracy, an AUC value of 0.7–0.9 indicates a medium accuracy, and an AUC value greater than 0.9 indicates a high accuracy.

in AHI values in the non-contact fiber-optic mattress group compared with the PSG group, indicating that although the non-contact fiber-optic mattress deviates from the PSG in terms of accuracy in assessing AHI values, it still has a high utility when used to assess specific degrees of OSAHS. PSG is expensive and inconvenient for long-term dynamic monitoring since it requires at least 7 h of overnight monitoring in a hospital and also requires the patient to wear numerous data sensors, seriously affecting sleep quality. Various OSAHS screening questionnaires such as the STOP Bang questionnaire and the Berlin questionnaire tend to be less accurate than monitoring (*Ha et al., 2014*), and their ability to diagnose OSAHS remains controversial (*Margallo et al., 2014*). Therefore, based on clinical and public demand, other simple and easy methods of OSAHS screening and diagnosis have been proposed (*Jagielski et al., 2022*, *Ding et al., 2022*).

Some studies use electromagnetic wave scanning reflection, but this monitoring method is easily affected by the external environment and the device hardware and algorithm are complex, making accurate monitoring difficult. There are also studies using pressure sensitive membrane sensors, but their sensitivity is relatively poor; the accuracy of physiological signal measurement is also poor and multi-level amplification is required on the circuit, which increases both circuit complexity and cost. This study used fiber-optic sensors, which are non-inductive and cause no physiological burden to the subject. The non-contact optical fiber mattress also has the advantages of non-sensory monitoring, data intelligence and long-term monitoring. A limited number of previous studies have been conducted to validate under-mattress sleep monitoring devices. *Tuominen et al. (2019)* compared the under-mattress Ballistocardit Beddit sleep tracker with PSG monitoring in 10 adults aged 18 to 30 years with a BMI of less than 30. This study showed that the two

devices correlated poorly in assessing indicators of sleep continuity. *Nagatomo et al. (2020)* compared the differences between a sub-mattress sensor, Nemuri SCAN, and PSG in 11 critically ill patients. The sensitivity and specificity of Nemuri SCAN compared to PSG were 90% and 39%, respectively; although the specificity was low, the sensitivity was high, which may be due to the limited ability of the sub-mattress sensor to recognize respiratory movements in critically ill patients with low activity levels (*Martinot et al., 2023*). In contrast, the body movement index of the non-contact optical fiber mattress group in this study was significantly higher than that of the PSG group, which may be because the non-contact optical fiber mattress uses fiber optics to sense respiratory motion, which is more sensitive and therefore senses more body movements than PSG monitoring.

This study compared sleep monitoring metrics of the non-contact optical fiber mattress with those from PSG monitoring. The results showed that mean heart rate, tachycardia index, bradycardia index, longest time of apnea, mean time of apnea, percentage of apnea to total sleep time, longest time of hypopnea, mean time of hypopnea and percentage of hypopnea to total sleep time were highly consistent between the two monitoring types. A Pearson correlation analysis showed that the AHI values of the non-contact optical fiber mattress were highly positively correlated with the AHI values of PSG. The ROC curve analysis also confirmed that the non-contact optical fiber mattress had a high specificity and sensitivity for the diagnosis of OSAHS, making it an effective device for assessing sleep breathing indicators and a reliable tool for OSAHS screening.

Sleep is vital to health (*Ramar et al., 2021*). *Buysse (2014)* defines sleep health as a multidimensional model that adapts to the needs of the body, society, environment and promotes physical and mental health. PSG monitoring in hospitals is still the gold standard for assessing sleep quality and diagnosing sleep apnea syndrome. However, PSG monitoring is expensive and requires professional personnel for operation, making PSG inaccessible for sleep health assessments in the general population. Quality sleep is defined not only by a lack of sleep disorders, but also based on sleep indicators such as sleep time. *Buysse*'s *(2014)* view on good sleep health emphasizes the following dimensions of sleep: subjective satisfaction, appropriate sleep time, sufficient sleep time, high sleep efficiency and continuous vigilance when awake. Due to the rapid development of artificial intelligence (AI), especially machine learning algorithms, home sleep monitoring equipment based on AI has become easy to use by the general population (*Khosla et al., 2018*). Thus, comprehensive and longitudinal sleep monitoring can be realized by all consumers with monitoring devices that can automatically output the sleep monitoring results to a computer or phone. Validation studies evaluating the performance of such household sleep monitoring devices based on PSG are crucial to verifying the accuracy of the monitoring results (*Lim et al., 2020*). The non-contact optical fiber mattress is one of the few sleep monitoring devices that is placed under the mattress during sleep, measuring apnea and hypopnea events in a non-intrusive way. The weak force of respiratory movement then acts on the fiber, resulting in micro-bending of the fiber, which leads to changes in light intensity. The light detector then adaptively detects these small changes in light intensity and uses them to measure physiological parameters such as heart rate, respiratory rate and body movement. The results of this study show that the non-contact

optical fiber mattress performs well in overnight sleep monitoring with a high specificity and sensitivity for OSAHS diagnosis.

The non-contact optical fiber mattress uses highly sensitive fiber-optic sensors to record forces on the mattress, including forces from somatic movements, respiratory movements and heartbeat and snoring vibrations, and then uses artificial intelligence to process those signals and calculate sleep monitoring results using computerized algorithms for OSAHS disease screening and diagnosis. In addition to being able to use it in the home, the non-contact optical fiber mattress also has the benefits of being accessible to the general public and capable of long-term dynamic monitoring and not requiring professional medical staff for monitoring. These advantages make this device beneficial in both health monitoring and chronic disease management.

This study also has some shortcomings, such as a small sample size and no obese participants included in the study. Future studies will try to expand the study population to provide more reliable experimental data for the diagnosis and treatment of OSAHS. The intelligent evaluation of sleep stage and respiratory variability of the non-contact optical fiber mattress is still being explored, and further studies are needed to verify the results of this study.

## LIST OF ABBREVIATIONS

| | |
|---|---|
| **PSG** | Polysomnography |
| **OSAHS** | Obstructive sleep apnea hypopnea syndrome |
| **AHI** | Apnea hypopnea index |
| **BMI** | Body mass index |

### Funding

This work was supported by the National key Research and Development Program of China (Project No. 2021YFC2500702) and the National Natural Science Foundation of China (No. 82270104, 82201268). Health Commission of Hubei Province scientific research project (WJ2023Z010); Young Doctors' Innovation and Development Program (HXQNJJ-2023-010); Moderate and Severe Asthma Diagnosis and Treatment of Scientific Research Project (Z001). The funders had no role in study design, data collection and analysis, decision to publish, or preparation of the manuscript.

### Grant Disclosures

The following grant information was disclosed by the authors:
National key Research and Development Program of China: 2021YFC2500702.
National Natural Science Foundation of China: 82270104, 82201268.
Health Commission of Hubei Province scientific research project: WJ2023Z010.
Young Doctors' Innovation and Development Program: HXQNJJ-2023-010.
Moderate and Severe Asthma Diagnosis and Treatment of Scientific Research Project: Z001.

## Competing Interests

The authors declare that they have no competing interests.

## Author Contributions

- Ling Zhou conceived and designed the experiments, performed the experiments, prepared figures and/or tables, and approved the final draft.
- Xiaoyan Zhu conceived and designed the experiments, performed the experiments, prepared figures and/or tables, and approved the final draft.
- Lu Liu conceived and designed the experiments, performed the experiments, analyzed the data, authored or reviewed drafts of the article, and approved the final draft.
- Lingling Wang performed the experiments, analyzed the data, authored or reviewed drafts of the article, and approved the final draft.
- Pengdou Zheng performed the experiments, analyzed the data, prepared figures and/or tables, and approved the final draft.
- Fengqin Zhang performed the experiments, analyzed the data, prepared figures and/or tables, and approved the final draft.
- Zhenyu Mao performed the experiments, prepared figures and/or tables, and approved the final draft.
- Guoliang Huang performed the experiments, authored or reviewed drafts of the article, and approved the final draft.
- Songlin Cheng performed the experiments, analyzed the data, authored or reviewed drafts of the article, and approved the final draft.
- Huiguo Liu performed the experiments, analyzed the data, authored or reviewed drafts of the article, and approved the final draft.
- Wei Liu conceived and designed the experiments, performed the experiments, authored or reviewed drafts of the article, and approved the final draft.

## Human Ethics

The following information was supplied relating to ethical approvals (*i.e.*, approving body and any reference numbers):

Tongji Hospital, Tongji Medical College, Huazhong University of Science and Technology granted Ethical approval to carry out the study within its facilities.

## Clinical Trial Ethics

The following information was supplied relating to ethical approvals (*i.e.*, approving body and any reference numbers):

Tongji Hospital, Tongji Medical College, Huazhong University of Science and Technology granted Ethical approval to carry out the study within its facilities.

## Data Availability

The raw measurements are available in the Supplemental File.

## Clinical Trial Registration

The following information was supplied regarding Clinical Trial registration:
TJ-IRB20221307.

## Supplemental Information

Supplemental information for this article can be found online at http://dx.doi.org/10.7717/peerj.17570#supplemental-information.

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
