# Peer review of "Recognition of apnea and hypopnea by non-contact optical fiber mattress and its application in the diagnosis of obstructive sleep apnea hypopnea syndrome: a retrospective study"

_PeerJ, doi:10.7717/peerj.17570_

## Round 0.1 · original submission · Minor Revisions

Dear Dr. Zhouzhou,

The reviewers have raised some minor revisions.

Please, revise the manuscript following the comments reported, so that the manuscript will be suitable for publication.

Thank you

**Language Note:** The review process has identified that the English language must be improved. PeerJ can provide language editing services - please contact us at [email protected] for pricing (be sure to provide your manuscript number and title). Alternatively, you should make your own arrangements to improve the language quality and provide details in your response letter. – PeerJ Staff

·

Basic reporting

Thanks to the publisher and the authors for allowing me to review this scientific article.
The article is well written and needs minor revisions in my opinion before being published.
In my opinion, the comparison between the two systems needs to be conceptually analyzed and reported in more studies before validating this method on a large scale, as one instrument calculates the values ​​indirectly, the other calculates them directly and the analyzed data can be altered and have important discrepancies, however the results of this study are really interesting.
These parameters used are very generic:
1) Sleep efficiency.
2) Total sleep time
I think they can be eliminated from the statistical analysis.

Experimental design

The experimental study was conducted excellently and rigorously, leading to a significant statistical analysis. it was well written.

Validity of the findings

I don't know how much impact this study has in terms of clinical practice, it is true that it is a less cumbersome non-invasive method compared to others, but it is also true that monitoring with the PSG instrument remains the "gold standard" for the diagnosis of sleep apnea and OSA/OHS. In my opinion, it is necessary to carry out studies on a larger population scale to consider the technology applicable on a large scale. My doubts remain about the validity in detecting central events, which in my opinion is not on a par with a tool like the PSG. However, it would be appropriate to specify this in the conclusions of the manuscript.

Reviewer 2 ·

Basic reporting

no comment

Experimental design

no comment

Validity of the findings

no comment

Reviewer 3 ·

Basic reporting

The article clearly states its objectives, which are to explore the diagnostic value of a non-contact optical fiber mattress for apnea and hypopnea and to compare it with traditional polysomnography (PSG).

Experimental design

The study’s design appears to be methodical. The comparison of non-contact optical fiber mattress readings with Gold standard PSG is a reasonable approach to validate the new technology.

Validity of the findings

The results indicate a clear distinction between the control group and OSAHS patients in terms of apnea and hypopnea frequency, as detected by PSG. The non-contact optical fiber mattress also reflected these differences, suggesting its potential as a diagnostic tool. after validity in a larger set of data

Additional comments

The article effectively highlights the limitations of PSG, such as cost, labor-intensiveness, and potential discomfort, which can impede timely diagnosis. The non-contact optical fiber mattress (ANVENUS SLEEP MONITOR) addresses these issues by offering a non-intrusive, comfortable, and convenient alternative for at-home diagnosis. Its ability to interface with a smartphone application for data analysis further enhances its user-friendliness and accessibility.
there are some minor grammatical errors for example
1.Line 27 “6 health” to “6 healthy”
2.Line 29 “ collected” to “Selected”
3.Line 32 remove the work big from “ big data”. Note check what big data means - this would not qualify as big data
4.Line 33 remove word intelligent-
5.Line 44 remove “detected by”

·

Basic reporting

The manuscript is well written apart from a few grammatical mistakes and English language changes that can be improved. There are some edits that can be made to the manuscript that are mentioned below

For the conclusion recommend editing the sentence for grammatical error. Below is an example.
The efficacy of non-contact optical fiber mattress monitoring for OSAHS is not significantly different from PSG. Its specificity for diagnosing OSAHS was 95%, and its sensitivity was 93%, indicating high accuracy in diagnosing OSAHS.

The title for Figure 2 can be non-contact optical fiber mattress waveform diagram in comparison with PSG waveform.

Experimental design

The research question is well defined and relevant. The study fills and identifies a knowledge gap to assess apnea and hypopnea as an alternative to PSG.

Validity of the findings

The authors have not mentioned in detail the limitations of the study in the discussion. For example, small size of population. Absence of obese population in the study. Who often have a higher prevalence of sleep apnea hypopnea. The authors can also mention the cost benefit ratio of PSG vs the non-contact
optical fiber mattress

The study and its findings are novel. The statistical analysis is appropriate for the study objectives. The conclusion of the study state the study findings correctly.

Additional comments

Line 59 remove space after (AUC=0.944) in “(AUC=0.944) , specificity(83.4%) and sensitivity(100%) for the diagnosis of hypopnea”

---

## Round 0.2 · accepted · Accept

Dear Dr. Zhouzhou,

The reviewers appreciated the revised manuscript and now your paper is suitable for publication.

·

Basic reporting

Ok with review that applied by the authors.

Experimental design

Ok with review that applied by the authors.

Validity of the findings

Ok with review that applied by the authors.

Additional comments

Ok with review that applied by the authors.

Reviewer 3 ·

Basic reporting

already submitted the reporting and no new comments are needed over and above the previous review.

Experimental design

already addressed

Validity of the findings

already addressed

Additional comments

none

·

Basic reporting

no comment. Article is edited and ready for publication

Experimental design

no comment

Validity of the findings

no comment

Additional comments

none